# Transportan Peptide Stimulates the Nanomaterial Internalization into Mammalian Cells in the Bystander Manner through Macropinocytosis

**DOI:** 10.3390/pharmaceutics13040552

**Published:** 2021-04-14

**Authors:** Yue-Xuan Li, Yushuang Wei, Rui Zhong, Ling Li, Hong-Bo Pang

**Affiliations:** 1Department of Pharmaceutics, University of Minnesota, Minneapolis, MN 55455, USA; li000613@umn.edu (Y.-X.L.); weiy@umn.edu (Y.W.); 2Department of Experimental and Clinical Pharmacology, University of Minnesota, Minneapolis, MN 55455, USA; zhong355@umn.edu (R.Z.); lil@umn.edu (L.L.)

**Keywords:** transportan, nanoparticles, bystander effect, macropinocytosis

## Abstract

Covalent coupling with cell-penetrating peptides (CPPs) has been a common strategy to facilitate the cell entry of nanomaterial and other macromolecules. Though efficient, this strategy requires chemical modifications on nanomaterials, which is not always desired for their applications. Recent studies on a few cationic CPPs have revealed that they can stimulate the cellular uptake of nanoparticles (NPs) simply via co-administration (bystander manner), which bypasses the requirement of chemical modification. In this study, we investigated the other classes of CPPs and discovered that transportan (TP) peptide, an amphiphilic CPP, also exhibited such bystander activities. When simply co-administered, TP peptide enabled the cells to engulf a variety of NPs, as well as common solute tracers, while these payloads had little or no ability to enter the cells by themselves. This result was validated in vitro and ex vivo, and TP peptide showed no physical interaction with co-administered NPs (bystander cargo). We further explored the cell entry mechanism for TP peptide and its bystander cargo, and showed that it was mediated by a receptor-dependent macropinocytosis process. Together, our findings improve the understanding of TP-assisted cell entry, and open up a new avenue to apply this peptide for nanomaterial delivery.

## 1. Introduction

Since the first nanoparticle (NP)-formulated drug, Doxil (liposomal formulation of doxorubicin), was proven for clinical use in the mid-1990s, nanomedicine research has attracted widespread interest and has led to an increasing number of clinical trials and clinically used drugs [1]. One prerequisite for NP-based therapeutics to be effective is the efficient entry into target cells [2]. The cell membrane comprises a lipid bilayer that presents a biological barrier to macromolecules, such as proteins and NPs [3]. A common strategy to increase NP internalization is to use cell-penetrating peptides (CPPs). CPPs are a family of short peptides typically consisting of 5–30 amino acid [4]. Based on their physical-chemical properties, CPPs are mainly categorized into three classes: cationic, amphiphilic and hydrophobic peptides [5]. These peptides explore two pathways for cell entry: direct penetration and endocytosis [4]. Direct penetration is usually energy-independent, can occur at low temperatures, and is resistant to endocytic inhibitors [4]. Endocytosis, on the other hand, is an energy-dependent process, which can be further classified as phagocytosis, clathrin- or caveolin-mediated endocytosis, and macropinocytosis [6]. While direct penetration has been reported for some CPPs (often at high concentrations) [7,8,9], it is broadly considered that most CPPs enter the cells through endocytosis [10,11,12].

Covalent coupling has been the mainstream strategy to facilitate the cell entry of macromolecule payloads (e.g., proteins and NPs) [13]. It has been also reported that endocytosis, initiated by CPP-receptor ligation, is the primary cell entry mechanism [4]. In spite of its effectiveness, this strategy requires additional chemical modifications on NPs [14]. To bypass this undesirable requirement, another co-administration strategy has been explored by us and others, in which CPPs and their payloads are simply co-administered, but not covalently linked (so-called “bystander” manner) [15,16,17,18]. The payloads (e.g., NPs or other types of macromolecules), termed bystander cargo, generally have a low or inability to enter the cells themselves during the time of observation, and have no physical/chemical interactions with CPPs. This effect is of potential value to drug delivery since it can circumvent the limited availability of target receptors on the cell surface and increase the specificity and therapeutic effects of approved drugs without additional modifications [14,19]. Kaplan et al. reported presumably the first evidence of CPP-driven bystander uptake, in which the cell entry of the transactivator of transcription (TAT) peptide increased the uptake of dextran, a solute tracer, in the bystander manner [15]. A more recent example arises from the studies of CendR peptides, which contain a positively charged R/KXXR/K motif on the C-terminus (*C*-end *R*ule, CendR) [16,17]. To be specific, iRGD, upon intravenous injection, was shown to increase the tumor accumulation and vascular penetration of a variety of bystander cargo types (small molecules, antibodies and NPs) [17]; LyP-1 and its linear truncated from tLyP-1, combined with doxorubicin hydrochloride, showed enhanced cytotoxicity in vitro and reduced tumor growth in vivo [19]; iNGR co-administration with doxorubicin also enhanced its anticancer efficacy [14]. Besides peptides, we also found that TAT-functionalized NPs (TAT-NPs) were able to stimulate the cellular uptake of NP-type cargo in the bystander manner [18]. It is an interesting cell biology question of how ligand-initiated endocytosis stimulates the internalization of bystander cargo from the nearby environment. In all these processes, macropinocytosis was proven as the primary endocytic mechanism for both CPPs and their bystander cargo [15,17,18]. While macropinocytosis has been traditionally regarded as a nonselective and receptor-independent process for fluid uptake, the aforementioned bystander uptake has been shown to rely on CPP-receptor interactions for initiation [18,20,21].

So far, the CPP-initiated bystander effect has only been observed within a few cationic CPPs [15,17,19]. Here, we aim to investigate whether other classes of CPPs can exhibit such bystander activities. Our results showed that transportan (TP), a 27-amino-acid amphiphilic CPP, also exhibits bystander activities towards an NP-type bystander cargo. TP consists of 12 functional amino acids from the neuropeptide galanin on the N-terminus and 14-amino-acid wasp venom peptide mastoparan on the C-terminus, connected via a lysine residue [22]. This peptide and its analogs are known for transporting a variety of cargo across the cell membrane, including proteins, siRNAs and NPs [23,24,25]. Our results here have provided a new dimension to use TP for cargo internalization, and insights into the cell entry mechanism for TP peptide and its bystander cargo.

## 2. Materials and Methods

### 2.1. Materials and Mice

CHO-K1 (CHO), CHO-pgs745 (CHO^pgs745^), 4T1, Hela, PPC1, PC3, NIH-3T3, H441, BxPC3, HUVEC, MCF10A and H1975 cells were purchased from the American Type Culture Collection (Manassas, VA, USA). H2122, A549 cells were generously provided by Dr. Garth Powis, MIA PaCa2 cells by Dr. Commisso, Sanford Burnham Prebys Medical Discovery Institute, San Diego, CA, USA; KPC cells by Dr. Andre Nel, University of California, Los Angeles, CA, USA; Ink cells by Dr. Douglas Hanahan, Swiss Federal Institute of Technology Lausanne, Lausanne, Switzerland. CHO, CHO^pgs745^ and A549 cells were cultured in Ham’s F-12K (Kaighn’s) Medium (F-12K, Thermo Scientific, Waltham, MA, USA); BxPC3, Hela, H1975, H441 and H2122 cells in RPMI 1640 (Thermo Scientific); 4T1, PPC1, PC3, NIH-3T3, MIA PaCa2, Ink and KPC cells in Dulbecco’s Modified Eagle’s Medium (DMEM, Hyclone, South Logan, UT, USA); HUVEC cells in EGM-2 Endothelial Cell Growth Medium (Lonza, Basel, Switzerland); MCF10A cells in Mammary Epithelial Cell Growth Medium (MEGM, Lonza). MEGM basal medium was supplemented with the additives provided within the MEGM kit (Cat# CC-3150, Lonza); all the other media were added with 10% fetal bovine serum (FBS), 1% penicillin/streptomycin and 2 mM L-glutamine.

Fluorescence dyes, CF555 and CF647 Succinimidyl Ester were purchased from Biotium (Fremont, CA, USA). Endocytic probes, Albumin from Bovine Serum (BSA, AF647 conjugate, Cat# A34758) and dextran (70,000 MW, Texas Red, Cat# D1830) were purchased from Invitrogen (Waltham, MA, USA). Surface coating chemicals, thiol-PEG2000-amine (Cat# JKA5143) and BSA (Cat# 12659) were obtained from Sigma (St. Louis, MO, USA). Transportan peptide (FAM-x-transportan, FAM-Ahx-GWTLNSAGYLLGKINLKALAALAKKIL) in l-configuration and other CPPs listed in Table 1 were all purchased from LifeTein, LLC (Somerset, NJ, USA). Receptor inhibition reagents, dextran sulfate (Cat# D4911), chondroitin sulfate (Cat# PHR1786), poly I (Cat# P4154), fucoidan (Cat# F8065) and heparin (Cat# H4784), were purchased from Sigma (St. Louis, MO, USA). Antibodies, Rab5 Rabbit mAb (Cat# 3547T) and Rab7 Rabbit mAb (Cat# 9367) were purchased from Cell Signaling Technology (Danvers, MA, USA).

Female BALB/c mice (7 weeks old) were purchased from Charles River (Wilmington, MA, USA). All animal studies were approved by University of Minnesota Institutional Animal Care and Use Committee (approval No. 1808-36261A dated 20 September 2018).

### 2.2. Preparation of Nanoparticles

NPs used in this study were synthesized as described in a previous study [18]. In brief, silver NPs (AgNPs) were synthesized via the reaction of AgNO_3_ and sodium citrate and then coated with HS-PEG2000-NH_2_. The complexes could be labeled by fluorescent NHS-CF555 and NHS-CF647 dyes to obtain AgNP-555 and AgNP-647 respectively. The mixture was then centrifuged and washed by PBST (1 × PBS + 0.005% Tween 20) buffer to remove excess dyes. The final concentration of AgNPs was adjusted to optical density (OD) 200 in PBST buffer.

Similar to AgNP synthesis, gold NPs (AuNPs) were synthesized via the reaction of HAuCl_4_ and sodium citrate and then coated with BSA [18]. Coated AuNPs were then centrifuged and washed by PBST buffer for 3 times to get rid of free BSA. The resulting complexes could be further labeled with NHS-CF647 to become fluorescent bystander cargo. The final concentration of AuNPs was OD 40 in PBST buffer.

Iron oxide NPs (IONPs) were synthesized by mixing ice cold reduced dextran (MW ~10,000 Da), FeCl_3_·6H_2_O, FeCl_2_·4H_2_O and NH_4_·OH together and then heated at 85 °C for 2 h [18]. After the solution cooled down to room temperature and was dialyzed (MWCO 50 kDa) for 72 h against double distilled water (DI water), IONPs could then be concentrated and aminated by adding 5 M NaOH, epichlorohydrin and 30% NH_4_·OH. The mixture was agitated for 24 h, dialyzed and concentrated using 50 kDa centrifugal filters (4000 *g*, 20 min). The resulting IONP colloid could be easily labeled with the NHS-CF647 dye. The final concentration of Fe in IONP solution was adjusted to 1 mg Fe/mL in PBS buffer.

The synthesis method of ZnS quantum dots (QDs) was also adapted from a previous protocol [26]. Briefly, 0.1 M Na_2_S was added to the degassed solution of zinc acetate and 3-mercaptopropionic acid at 50 °C. Ethanol was used to precipitate the QDs from the solution. The precipitate was then dissolved in DI water and mixed with HS-PEG2000-NH_2_ for amination. The amine groups on QDs could react with NHS-CF647 dyes to obtain bystander NPs. The final concentration of QDs was adjusted to 0.5 mg/mL in PBS buffer.

### 2.3. Dynamic Light Scatter (DLS)

The hydrodynamic size and zeta potential of NPs used in this study were determined in PBS buffer using a Nano ZS particle analyzer (Malvern Panalytical, Malvern, UK) from University of Minnesota Nano Center.

### 2.4. Cellular Uptake Study

NP internalization assay was performed as previously described [18]. Cells were grown in the 96-well plate (Cat# 2870-1002, Nest Scientific, Inc. Rahway, NJ, USA) containing 100 μL culture medium per well for 24 h. At about 80% confluency, 2 μL of NP solution was added into 100 μL serum-free DMEM medium per well for internalization. After 1-h incubation at 37 °C, etching buffer (a final concentration of 10 mM Na_2_S_2_O_3_ and 10 mM K_3_Fe(III)CN_6_ in DPBS buffer (Cat# SH30028.02, Hyclone, South Logan, UT, USA)) was added into the medium to eliminate extracellular AgNPs [27]. AuNPs attached to the cell surface could be removed by I_2_/KI solution (a final concentration of 0.34 mM I_2_ and 2.06 mM KI) [28]. Thereafter, cells were washed by DPBS buffer three times, detached using 0.25% Trypsin-EDTA (Cat# 25300-054, Gibco, Waltham, MA, USA) and finally fixed with 4% paraformaldehyde (PFA, Cat# sc-281692, Santa Cruz Biotech, Dallas, TX, USA). Internalized fluorescence intensity per cell was analyzed using flow cytometry on BD LSRFortessa (BD Biosciences, San Jose, CA, USA). The median fluorescence intensity of corresponding gated live single cells was used to evaluate the level of NP/macromolecule internalization into cells.

To quantify the cellular uptake of TP peptide, TP was added into the serum-free DMEM medium at the indicated concentrations. For bystander uptake quantification, bystander NPs (AgNPs, AuNPs, IONPs and QDs) and macromolecules (BSA and dextran at a final concentration of 0.2 mg/mL) were mixed with 10 μM TP peptide in the medium. To inhibit HS binding or scavenger receptors on the cell surface, soluble heparin (1000 μg/mL) or dextran/chondroitin sulfate (5 μg/mL) or poly I (50 μg/mL) or fucoidan (1500 μg/mL) was incubated together with 10 μM TP peptide and bystander cargo in the FBS-free medium.

### 2.5. Physical Interaction Assay

The protein binding study was performed according to a previous protocol [29]. TP peptide (50 μL, 5 μg/mL) was immobilized on the 96-well high-binding plate (Fisher Scientific, Cat# 0720039) at 4 °C overnight. After the peptide solution was aspirated, wells were washed with PBS. 1% BSA in PBS was then added and incubated at room temperature for 1 h to block the remaining binding sites. After washing with PBS three times, the wells were incubated with bystander cargo (100 μL serum-free DMEM per well) at room temperature for 1 h. The wells were then washed by PBS three times and fluorescence intensity of bound bystander cargo was measured by a microplate reader (Synergy H1, BioTek, Winooski, VT, USA). The CF647 signal in each well was measured with the setting: excitation 640 nm, emission 668 nm.

### 2.6. In Vitro Imaging

Cells were grown on glass chamber slides containing 0.05 × 10^6^ cells per chamber (eight chamber, Nunc, Scotts Valley, CA, USA) for 24 h. Cells were then washed by DPBS buffer once and incubated with AgNP-555 or AgNP-555 + TP peptide in serum-free DMEM medium at 37 °C. 1 h later, cells were etched and washed with PBS. To stain early/late endosome markers, cells were first treated with PBS containing 1% BSA and 0.1% Trition X100 (blocking buffer) at room temperature for 1 h. Cells were washed with PBS and then incubated with primary antibodies diluted (1:200) in blocking buffer at 4 °C overnight, followed by appropriate secondary antibodies diluted (1:200) in blocking buffer at room temperature for 1 h. After PBS washing, cells were fixed with 4% PFA. The chamber slides were mounted in DAPI-containing mounting medium (Vector Laboratories, Burlingame, CA, USA) by coverslips (Leica, Buffalo Grove, IL, USA, Cat# 3800145ACS). Nikon C2 confocal (Melville, NY, USA) was used here for imaging.

### 2.7. In Vitro EXT Knockdown Experiment

Hela cells were cultured in the 12-well plate (Cat# 712001, Nest Scientific, Inc.) containing 1 mL culture medium per well. At about 50% confluency, EXT siRNAs were mixed with lipofectamine RNAiMAX Reagent (Cat# 13778, Fisher, Waltham, MA, USA) and added into the well according to the manufacturer’s protocol. Six hours after transfection, the medium was replaced. After 48 h, cells were harvested for RNA isolation according to the supplier’s protocol (Cat# R2061, Zymo Research, Irvine, CA, USA). RNA samples were then converted to cDNA by reverse transcription reaction (Cat# 4368814, Fisher). Reverse-transcription polymerase chain reaction was performed in a total volume of 20 μL containing SYBR green master mix (Cat# A25776, Fisher), cDNA and specific primer pairs. Primers used to assess the level of suppression by siRNA were designed with PrimerBank and were purchased from Integrated DNA Technologies. Target cDNA was amplified using the Applied Biosystems as follows: initial denaturation at 95 °C for 10 min, followed by 42 cycles of 30 s at 95 °C, 1 min at 55 °C for annealing, and 40 s at 72 °C for elongation. The amplification was ended with a melting curve starting after 40 cycles. Gene expression was calculated using GAPDH as a housekeeping gene with the ΔΔCt calculation.

To investigate the effect of EXT-silencing on TP-induced bystander uptake, 48 h after transfection, Hela cells were incubated with 10 μM TP peptide and bystander cargo (AgNPs, AuNPs and IONPs) in the FBS-free medium for 1 h. Cells were then etched, washed with PBS and subjected to flow cytometry for fluorescence analysis.

### 2.8. Transmission Electron Microscopy (TEM)

CHO cells were incubated with Au50 or Au50 + TP peptide in serum-free DMEM medium at 37 °C for 2 h. Then cells were processed at the Characterization Facility of University of Minnesota and imaged by FEI Tecnai Spirit Bio-Twin TEM (FEI, Hillsboro, OR, USA) as previously described [18].

### 2.9. Ex Vivo Tissue Uptake

For mouse 4T1 breast tumor production, a million tumor cells suspended in DPBS were transplanted into the mammary fat pad of female BALB/c mice orthotopically, as described previously [29]. After 14 days, mice were sacrificed under deep anesthesia and tumors were collected. Live 4T1 tumor slices were then sectioned at the thickness of 200 μm using a Leica VT1000P vibratome. Tumor slices were cultured in the 12-well plate (VWR, West Chester, PA, USA, Cat# 82050-930) and incubated with AgNP-555 or AgNP-555 + TP peptide in the FBS-free DMEM medium at 37 °C for 2 h. After incubation, the slices were etched, washed by DPBS, fixed with 4% PFA and mounted with the DAPI-containing mounting medium by coverslips. Nikon C2 confocal (Melville, NY, USA) was used to examine the tumor slices.

### 2.10. Statistical Analysis

All data were presented as mean ± standard deviation (s.d.). Each experiment was repeated at least three times. Data analysis was performed using a one-way ANOVA with Turkey’s multiple comparison test, unless otherwise indicated. Graphpad prism was used for statistical computing. Calculated *p* values at <0.05 were considered significant.

## 3. Results

### 3.1. Bystander Activities of CPP Classes

As mentioned above, bystander activities have been only seen with cationic CPPs [15,17,19]. Here, we synthesized a list of example CPPs covering all three classes (Table 1). We used silver nanoparticles (AgNPs) as the model bystander NPs for the initial screening. AgNPs were chosen due to the ease of visualizing and quantifying their internalization by fluorescence measurements. Moreover, an etching method has been developed to rapidly dissolve the extracellular AgNPs, while leaving those internalized unharmed, which allowed us to accurately monitor the internalized ones [27]. As shown in Table 1, another cationic peptide, R9, exhibited a similar bystander activity as TAT. Except for TP, we observed little bystander activities for hydrophobic and amphiphilic ones. Therefore, we focused on TP peptide in the following studies.

### 3.2. Cellular Uptake of TP and Bystander AgNPs

Previous studies on TP used various experimental conditions (e.g., cell type, peptide concentration, cargo type, etc.) [23,24,25,30]. Thus, we first optimized the cellular uptake of TP in a variety of cell lines and experiment conditions. Cell line studies showed the internalization of TP depended on the cell type (Figure 1A). H1975, Hela and CHO-K1 (CHO) cells exhibited the highest activity of engulfing TP, thus being used as the primary cell lines in the following studies. It is noteworthy that TP had lower capacity to enter CHO-pgs745, a CHO mutant cell line without heparan sulfate proteoglycans (HSPGs) expression [20], compared to CHO wild type. This result suggested that TP uptake depended on its interactions with HSPGs on the cell surface.

Next, we tested a titrating concentration of TP peptide to see the optimal one for its bystander activity. Using a CHO cell line and bystander AgNPs, we found that the bystander activity of TP was peaked from 10 to 20 µM (Figure 1B). In the subsequent studies, we used 10 μM as the optimal TP concentration. Besides flow cytometry, we also used confocal imaging to observe the internalization of TP and bystander cargo (Figure 1C). In the absence of TP peptide, no AgNP signal was observed, consistent with the low fluorescence intensity in cells treated with AgNPs alone, which was quantified by flow cytometry. When co-administered with TP peptide, a much higher level of AgNP uptake was observed together with TP peptide. Inside the cells, we observed some colocalization between these two. Overall, these results demonstrated that TP co-administration greatly increased the cellular internalization of AgNPs.

### 3.3. TP-Induced Bystander Uptake of NPs

Next, we tested the TP-induced bystander activities towards other cargo types, especially NPs. Bystander NPs used in this study and their physicochemical properties were summarized in Table 2. Particle size distribution profiles were showed in Appendix A. Different surface-coating strategies were used, and the average particle sizes varied from 30 to 100 nm. The polydispersity intensity (PDI) is an important indicator of particle quality with respect to the size distribution [31]. Here, PDI of our nanoparticle formulations were all about 0.3 and below, which was considered to be acceptable and indicated a homogenous population of NPs [31]. Besides NPs, fluid markers (or solute tracers), such as BSA (albumin from bovine serum) and Dextran (MW 70 kDa), were also used in the uptake experiments [18]. To avoid cell-derived variations, we used three cell lines in this study. Though unable to enter the cells efficiently by themselves, the uptake of AgNPs, AuNPs and IONPs was significantly increased when co-administered with TP peptides in all tested cell lines (Figure 2A). TP exhibited bystander activities towards QDs in CHO and H1975 cells, but not in Hela cells. We also observed the bystander activity of TP towards Dextran in all cell lines, and towards BSA in CHO cells but not the other two. The reason for this may be that BSA uptake by H1975 and HeLa was already efficient enough without TP, and thus it was difficult to observe the bystander activity. 70 kDa dextran is commonly regarded as a macropinocytosis tracer since its relatively large size excludes the possibility of being engulfed by other endocytic pathways [19]. Therefore, this result also suggested that macropinocytosis was stimulated by a TP peptide. Other than tumor cell lines and fibroblasts, we also examined the TP-induced bystander effect in human umbilical vein endothelial cells (HUVECs), as shown in Appendix A. Although the uptake of AgNPs, AuNPs and IONPs was significantly increased with TP co-administration, bystander activities towards BSA and Dextran did not achieve statistical significance, mainly due to the low level of TP uptake in HUVECs as compared to other cell lines (Figure 1A).

One prerequisite for NPs to be qualified as bystander cargo is that they have no physical interaction with TP peptides [18]. We next set out to rule out the possibility that the nanomaterial bystander effect was an artifact due to the violation of the above. We coated TP peptides onto the protein-binding plate and incubated these wells with 1% BSA to block the remaining binding sites on the plate, according to a previous protocol [29]. A variety of bystander NPs (AgNPs, AuNPs, IONPs and QDs) in serum-free medium at cell treatment concentrations were added into the wells. After washing to remove unbound NPs, we observed low or no signals for these bystander NPs bound to the plate, as shown in Figure 2B. Collectively, these results showed that these bystander NPs did not interact physically with TP peptides, satisfying the prerequisite of bystander uptake.

### 3.4. TP-Induced Bystander Uptake Mediated by Receptor-Dependent Macropinocytosis

To understand the mechanism of TP-induced bystander uptake, we first tested whether TP uptake was a receptor-dependent process. Despite having been discovered for decades, the the cellular receptors for TP remain unclear [30]. Here, our results with CHO-pgs745 cells indicated an involvement of HSPGs in this process (Figure 1A). Another candidate is scavenger receptors, due to their roles in the cellular uptake of other CPPs [32]. As shown in Figure 3A, we observed that TP uptake was significantly decreased in cells treated with a scavenger receptor inhibitor, dextran sulfate, but not the negative control chondroitin sulfate, indicating scavenger receptors were involved in TP internalization. Similarly, soluble heparan sulfate was shown to significantly reduce the TP internalization, indicating that TP-HSPG interaction was required for TP uptake into cells (Figure 3B). Next, we would like to verify whether these receptors were also involved in TP-induced bystander uptake of NPs. A variety of bystander NPs (AgNP, AuNP and IONP) were used to co-incubate with TP peptide alone, or with dextran sulfate or heparan sulfate. Our results showed that TP-induced bystander uptake was significantly decreased by these inhibitors as well (Figure 3C). Scavenger receptors are a large group of structurally unrelated molecules known to mediate the endocytosis of certain polyanionic ligands, and moreover, multiple types of scavenger receptors with overlapping functions may participate together in pattern recognition [33]. Furthermore, based on gene expression database, at least 17 members of scavenger receptor family are expressed in Hela cells (Appendix A). Therefore, instead of exploring the contributions of individual genes, we used additional pharmacological reagents to further validate the involvement of scavenger receptors as a whole in TP-induced bystander uptake. Poly I and Fucoidan, two known inhibitors of the scavenger-receptor family [34], reduced the cellular uptake of TP and bystander NPs (Appendix A). On the other hand, HSPGs are a large heterogeneous group of heavily glycosylated proteins but they all contain heparan sulfate chains on the proteoglycan backbone [35]. The members of the exostosin family (EXT) are involved in heparan sulfate chain initiation and elongation [36]. To further validate the role of HSPGs, we used small interfering RNAs (siRNAs) against EXT1 and EXT2 to investigate the effect of disrupted HS modifications on TP-induced bystander uptake (Appendix A). Knocking down individual EXTs did not exhibit any effect on TP uptake, suggesting that there may be redundancy in terms of EXT1 and EXT2 functions (Appendix A). In contrast, knocking down both genes simultaneously resulted in a significant reduction of cellular uptake of both TP peptide and bystander NPs (Appendix A). Together, the internalization of TP peptide and its bystander cargo relied on cellular receptors, including scavenger receptors and HSPGs. The exact mechanism of TP interactions with these receptors remains to be further characterized.

To confirm the endocytic pathway that TP invokes to engulf bystander NPs, we used transmission electron microscopy (TEM) to visualize the endocytic process at the ultrastructural level. 50 nm gold nanoparticles (Au50) were used as bystander NPs here and they showed little ability to enter the cells themselves (Figure 4a). When co-administered with TP peptide, Au50 were internalized in the macropinosome-like (>200 nm in diameter) vacuoles (Figure 4b). Inside the cells, Au50 appeared in early endosomes (Figure 4c) and late endosomes/multivesicular bodies (Figure 4d). To confirm the intracellular trafficking of the endocytosed NPs, green fluorescence of TP peptide and red signal in AgNPs with two specific protein markers of endosomes, Ras-related proteins 5 and 7 (Rab5 and Rab7) were observed, indicating the formation of early and late endosomes respectively (Figure 4e,f). 3D sectioning images of bystander AgNPs and early/late endosomes were taken on the confocal microscope layer by layer and then reconstructed (see Appendix A). Results consistently showed that the corresponding particle-containing vesicles were characteristic of early and late endosomes. These results, along with the Dextran uptake result (Figure 2A), supported the notion that TP peptide initiated a receptor-dependent, macropinocytosis-like process for bystander uptake.

### 3.5. TP-Induced Bystander Uptake under Physiological Conditions

Last, we confirmed the above observation under physiological conditions. Live tumor slices from 4T1 mouse breast cancer were previously used as physiologically relevant samples for such a purpose [18]. Here, we prepared such tumor slices to quantify TP internalization and its bystander activity. GGS peptide (GGSGGSKG) was used as a peptide control, which was unable to enter the cell [29]. AgNPs, used as the bystander NPs, exhibited no or little ability to enter these tumor slices when used alone or together with GGS peptide. In contrast, TP peptide was able to efficiently bind and internalize into these tumor slices itself, and significantly increase AgNP uptake as well (Figure 5A,B). Together, we proved that TP peptide was able to induce the bystander uptake of NPs in physiologically relevant samples.

## 4. Discussion

CPPs have been widely reported to efficiently deliver nanomaterial and other types of cargo into cells in vitro and in vivo [4,5,6,13]. The most commonly used method is to chemically conjugate CPPs with NPs to invoke endocytosis processes for the cell entry [16,20]. Alternatively, a few studies have reported that some cationic CPPs have the ability to induce NP uptake in the bystander manner [19]. In this work, we extend the bystander phenomenon to another class of CPP by demonstrating that TP, a widely used amphiphilic CPP, can also stimulate the cellular uptake of NPs and other cargo types in the bystander manner. This result has been seen in a variety of cell lines, NP types, and ex vivo. We have also confirmed that this process is mediated by a receptor-dependent macropinocytosis process. Overall, our results have broadened the applicability of bystander phenomenon and provided insights in TP-mediated cellular internalization.

The bystander uptake mechanism, where a cell-penetrating ligand and inert payloads are simply co-administered for cellular uptake, has some unique advantages over the traditional strategy of covalent coupling. First, no chemical modification is needed on CPPs and payloads (e.g., NPs). Second, this strategy may circumvent the limitation of receptor availability on the cell surface [14]. This benefit has been best showcased by a tumor-specific CendR peptide, iRGD [17]. Upon systemic administration, iRGD selectively recognizes tumor vasculatures and penetrates deep into the extravascular regions along with a variety of co-administered drug types, ranging from small molecules to NPs [14]. This property allows iRGD to increase the tumor accumulation and antitumor efficacy of clinically used drugs without any chemical modifications. Despite these advantages, nonselective engulfment of materials (e.g., plasma components, co-administered drugs) into cells and tissues may induce adverse effects. Clinical trials are ongoing to determine any potential adverse effect of iRGD when used in the co-administration manner. On the other hand, the bystander activity of iRGD is largely limited to the tumor tissue where its targets mainly reside [14,17]. Therefore, improving the targeting specificity may be one way to help attenuate or eliminate adverse effects induced by nonselective uptake.

On the other hand, the bystander uptake activity has been only seen with a few cationic peptides [19]. Here, we have made representative CPPs from each class and tested their bystander activities to see whether this phenomenon is specific to cationic CPPs. TP has been shown to be the only non-cationic CPP that exhibits a measurable bystander activity. Using multiple cell lines, we have validated that TP peptide can stimulate the cellular uptake of multiple types of bystander NPs. TP peptide has exhibited no physical interaction with all bystander NPs used in this study, confirming that the observed enhancement is not an artifact. This bystander activity, despite varying with NP and cell types, is similar to previous reports [17,18]. The bystander activity of TP peptide also applies to solute tracers, Dextran and BSA, which is similar to TAT peptide but not TAT-functionalized NPs [18]. This result highlights the difference of CPP activity when used as free peptides versus being polymerized on the NP surface. Last but not least, we have used live tumor slices to confirm the TP-induced bystander uptake under physiological conditions. Overall, our studies have illustrated that nanomaterial entry into cells in a TP-induced bystander manner exists both in vitro and under physiological conditions.

Although TP has been widely used in vitro, its exact cell entry mechanism remains elusive. First, both direct translocation and endocytosis have been reported to be involved [22,37,38,39]. Here, we have optimized the cell entry of TP in a wide variety of cell lines, and the concentration used to stimulate the bystander uptake. The CPP concentration has been reported to influence the mechanism for its internalization [7]. For example, direct translocation into cells happens at relatively high concentrations (>50 μM). Here, we have rather found that a relatively low concentration (10 μM) is best suited for TP to induce the bystander uptake. Second, inhibitor studies have revealed that HSPGs and scavenger receptors are likely to be the cellular receptors for TP and are required for efficient internalization of TP peptide and its bystander payloads. Pathologically, scavenger receptors and HSPGs are both involved in absorption of the virus to the cell surface, acting together to mediate viral infection [40]. Moreover, they have also been reported to function as endocytic receptors for internalizing a variety of cargo types, including nucleic acids, NPs and antibodies [34,35]. Here, we have identified HSPGs and scavenger receptors as two types of membrane proteins primarily responsible for facilitating the cellular uptake of TP peptide and bystander NPs. However, as there are no reported direct molecular targets for TP peptides [22,37,38], we cannot exclude the possibility that TP-induced bystander uptake involves more than just an interaction with HSPGs and scavenger receptors. Our studies may aid the understanding of endocytic receptors that bind TP peptides and provide insights into potential cell surface targets for NP delivery. Finally, TEM studies have showed that TP initiates a macropinocytosis-like process to engulf bystander NPs. Traditionally, macropinocytosis has been considered as a receptor-independent and nonselective pathway for cells to internalize extracellular fluids and solutes contained within the large-sized membrane protrusions (>200 nm in diameter) [19]. Our recent studies have showed a macropinocytosis-like process initiated by ligand-receptor binding, which facilitates the cellular uptake of bystander NPs co-administered with CPPs [18,20]. Here, we have showed that this receptor-mediated macropinocytosis can also be invoked by TP, a non-cationic CPP. Together, these results provide new insights into TP-induced bystander uptake and the cell entry mechanism of TP in general.

Although many CPPs have been recently reported to assist drug delivery in vivo, most of the used CPPs belong to the cationic class, such as R9 and TAT [41]. Amphiphilic peptides, such as TP, has achieved limited success in clinical settings [41]. This issue may be attributed to the routes and methods of administration, the payload types, the target cell/tissue types as well as other factors [35]. Our current study mainly concerns the bystander activity of TP peptides during its entry into cells, and the in vivo applications are beyond our scope. However, this bystander activity may provide an alternative way for TP applications in drug delivery, in vitro or in vivo. HSPGs and scavenger receptors are expressed in a wide variety of cell and tissue types, and thus this phenomenon may be applicable under many conditions. The co-dependence on both receptor types indicates the possibility of a synergistic effect of them during TP uptake and TP-induced bystander uptake. Additionally, our data show that TP-induced bystander activity is more prominent for some NP types than others, which provides insights into the choice of cargo types for TP-facilitated delivery.

## 5. Conclusions

In summary, we show here that TP, an amphiphilic CPP, can increase the cellular uptake of nanomaterial cargo in the bystander manner. Following TP co-administration, a variety of NP-type bystander cargo and fluid markers can efficiently internalize into the cells in vitro and ex vivo. We have discovered that the NP entry through TP-induced bystander uptake is through a specialized endocytic pathway, the receptor-dependent macropinocytosis. Together, our findings present a useful strategy that can lead to more efficient drug delivery development. Future directions may include determining the physiological conditions best-suited for TP-mediated delivery in the bystander manner, and exploring its utilization in improving the intracellular delivery and therapeutic efficacy of NP-formulated drugs. Additionally, it remains to be addressed in regard to the optimal physiochemical properties of bystander NPs (size, shape, composition, etc.) for TP-mediated bystander uptake. Further investigations are also needed to fully address the roles and interactions of HSPGs and scavenger receptors in TP-mediated intracellular delivery and bystander uptake. These investigations may advance our understanding of TP cell entry and TP-mediated bystander uptake, and provide a new way to increase the intracellular delivery efficiency of NPs as well as other macromolecule types.

## Figures and Tables

**Figure 1 pharmaceutics-13-00552-f001:**
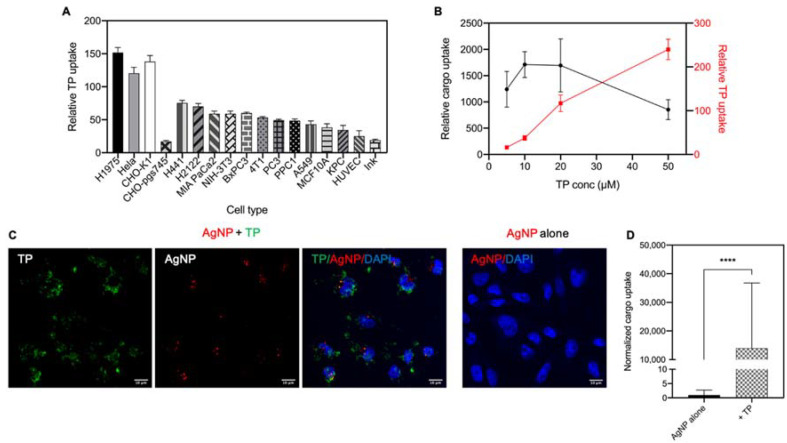
The optimization of TP internalization and induced bystander uptake. (**A**) Quantification of TP peptide uptake in various cell lines. Cells were treated with 10 μM of FAM-labeled TP peptide (FAM-TP) at 37 °C for 1 h. After washing, the fluorescence intensity of TP peptide per sample was detected and measured by flow cytometry. The median fluorescence value was normalized to that of corresponding cells alone (*y*-axis). Error bars, mean ± standard deviation (s.d.) (*n* = 3). Statistically significant difference between various cell lines was determined by one-way ANOVA. F (15, 32) = 180.5, *p* < 0.0001. The uptake in CHO-pgs745 was significantly lower in comparison with that in CHO-K1 when two tailed Student’s t-test was performed (*p* < 0.0001). (**B**) Quantification of TP and bystander AgNP uptake by CHO cells at different concentrations of TP. As described in the Methods, CHO cells were incubated with AgNP and TP peptide together at the indicated concentrations (5, 10, 20, 50 μM). After washing, the fluorescence intensity per sample was quantified by flow cytometry. The median fluorescence intensity of internalized bystander AgNP was normalized to that of cells incubated with AgNP alone (left *y*-axis, black). Error bars, mean ± standard deviation (s.d.) (*n* = 3). Relative AgNP uptake was not significantly different from each other using one-way ANOVA with Tukey’s multiple comparisons test. Left *y*-axis, relative AgNP uptake (black). Right *y*-axis, relative TP uptake (red). (**C**) Confocal images of CHO cells after incubation with FAM-TP (green) and bystander AgNP (AgNP-555, red). Representative images were shown here. Scale bars, 10 μm. (**D**) Quantification of bystander AgNP uptake by CHO cells. Mean fluorescence intensity of AgNP-555 obtained from three independent experiments (*n* = 3) through Image J software was normalized to that of AgNP alone group (*y*-axis). Values shown here were mean ± s.d. There was a statistically significant difference between AgNP alone group and TP group as determined by two tailed Student’s t-test. **** *p* < 0.0001.

**Figure 2 pharmaceutics-13-00552-f002:**
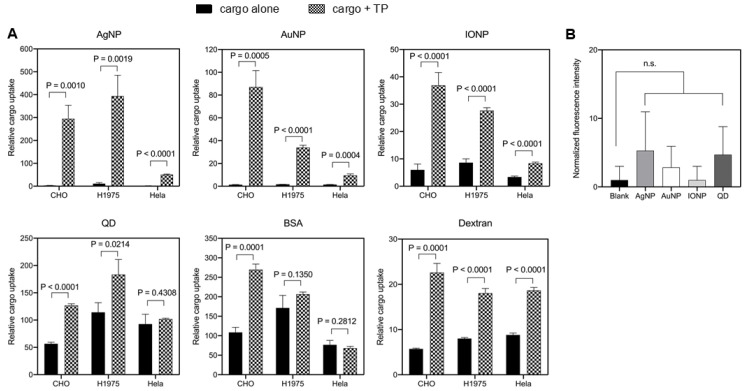
TP-induced bystander uptake. (**A**) Quantification of bystander cargo uptake by multiple cell lines. Bystander cargo (AgNP, AuNP, IONP, QD, BSA and dextran) was either mixed with indicated cell lines alone or together with TP peptide at 37 °C for 1 h, as described in the Methods. After etching or washing, the fluorescence intensity of internalized cargo was quantified by flow cytometry and normalized to that of corresponding cells alone (*y*-axis). Error bars, mean ± standard deviation (s.d.) (*n* = 3). Two tailed Student’s t-test was performed. *p* value was indicated and was considered significant when *p* < 0.05. (**B**) Physical interaction study between TP and NPs. The plate binding assay was performed as descried in the Methods. Fluorescence intensity of bound NPs (AgNPs, AuNPs, IONPs and QDs) in each well was normalized to that of wells incubated with medium alone (blank). Data shown here were mean ± s.d. of at least three independent experiments and were analyzed using one-way ANOVA with Tukey’s multiple comparisons test. n.s., not significant.

**Figure 3 pharmaceutics-13-00552-f003:**
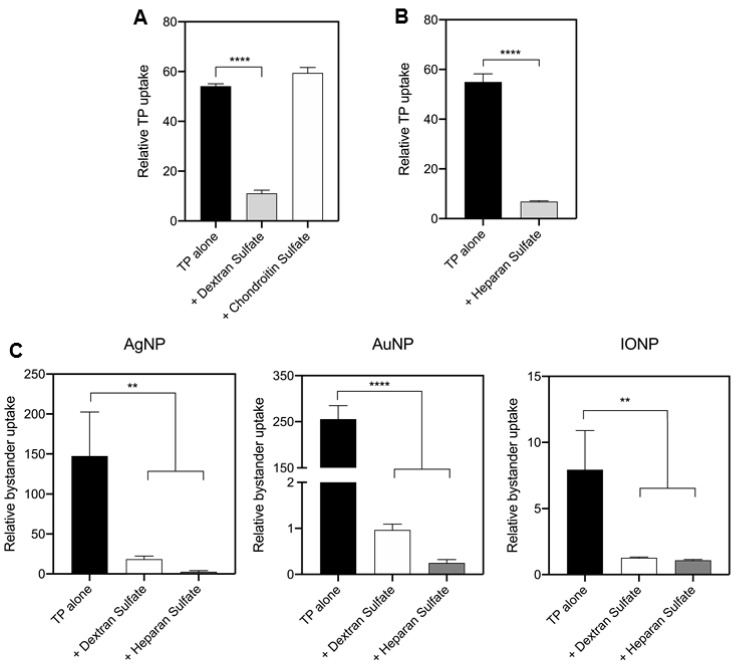
Receptor dependence. (**A**) Blockage of TP uptake by a scavenger receptor inhibitor. CHO cells were incubated with 5 μg/mL dextran sulfate or chondroitin sulfate together with TP peptide for 1 h, as described in the Methods. Cells were then washed, detached and subjected to flow cytometry. Fluorescence intensity of internalized TP peptide was normalized to that of cells alone (*y*-axis). Error bars, mean ± standard deviation (s.d.) (*n* = 3). Two tailed Student’s *t*-test was performed. **** *p* < 0.0001 in comparison with the TP alone group. (**B**) Blockage of TP uptake by heparan sulfate inhibition. CHO cells were incubated with 1000 μg/mL soluble heparin together with TP peptide for 1 h, as described in the Methods. Flow cytometry was used to quantify the fluorescence intensity of internalized TP peptide which was further normalized to that of cells alone (*y*-axis). Error bars, mean ± standard deviation (s.d.) (*n* = 3). Two tailed Student’s *t*-test was performed. **** *p* < 0.0001 in comparison with the TP alone group. (**C**) Blockage of NP bystander uptake by heparan sulfate and a scavenger receptor inhibitor. As described in the Methods, CHO cells were incubated with indicated bystander NPs (AgNPs, AuNPs and IONPs) + TP peptide alone, or together with dextran sulfate or heparan sulfate. Fluorescence intensity of bystander NPs per sample was quantified by flow cytometry and normalized to that of cells with bystander NPs alone (*y*-axis). Data were presented as mean ± s.d. (*n* = 3) and analyzed using one-way ANOVA with Tukey’s multiple comparisons test. ** *p* < 0.01, **** *p* < 0.0001 in comparison with the TP alone group.

**Figure 4 pharmaceutics-13-00552-f004:**
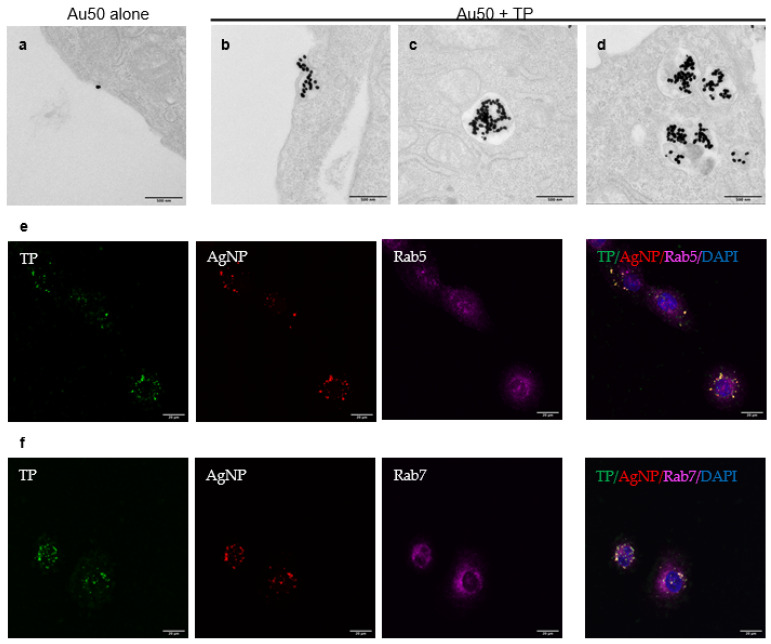
Imaging of endocytic structures for bystander uptake. CHO cells were incubated with Au50 alone (**a**) or Au50 + TP (**b**–**d**) in the serum-free DMEM medium for 2 h before fixation and being processed for TEM imaging. Ultrastructurally, AuNPs were found in macropinosome-like endocytic vacuoles (>200 nm in diameter) (**b**), early endosomes (**c**) and late endosomal multivesicular bodies (**d**). Presentative images were shown here. Scale bars, 500 nm. (**e**,**f**) Colocalization of bystander AgNP with endosome markers as revealed by confocal imaging. Hela cells were incubated with FAM-TP (green) and bystander AgNP (AgNP-555, red) for 1 h. Endosome markers (magenta) were stained by immunofluorescence, as described in the Methods. Bystander AgNPs were caputured in Rab5-localized early endosomes (**e**) and Rab7-localized late endosomes (**f**). Scale bars, 20 μm.

**Figure 5 pharmaceutics-13-00552-f005:**
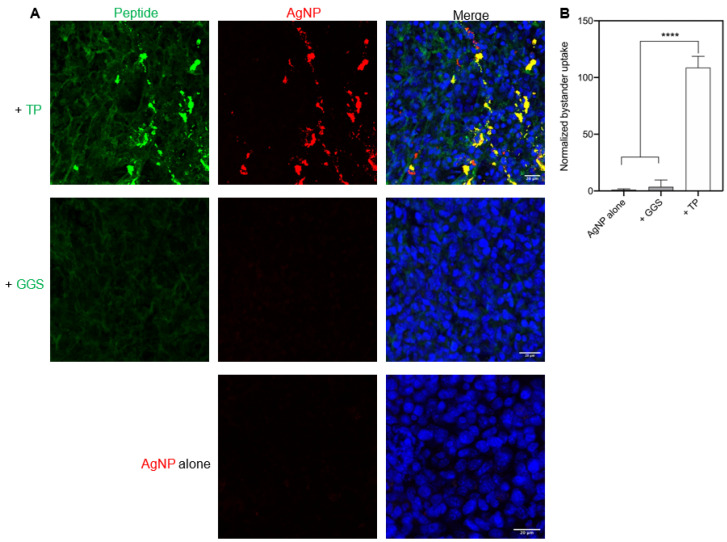
TP-induced bystander uptake under physiological conditions. (**A**) Confocal microscopy images of bystander AgNP uptake by live tumor slices. AgNPs (AgNP-555, red) were incubated with live 4T1 tumor slices either alone, or with TP or GGS peptide, as described in the Methods. After 2-h incubation, slices were etched, stained with DAPI (blue), and then subjected to imaging. GGS was a control peptide. Representative images were shown here. Scale bars, 20 μm. (**B**) Quantification of AgNP uptake by live tumor slices. Mean fluorescence intensity of bystander cargo (AgNP-555) obtained from three independent experiments (*n* = 3) through Image J software was normalized to that of AgNP alone group (*y*-axis). Values shown here were mean ± s.d. There was a statistically significant difference between TP group and other groups as determined by one-way ANOVA with Tukey’s multiple comparisons test. **** *p* < 0.0001.

**Table 1 pharmaceutics-13-00552-t001:** Example CPPs and their bystander activities.

CPPs	Classes	Sequences	Normalized AgNP Uptake
RPAR	Cationic	RPARPAR	1.01 ± 0.01
TAT	Cationic	YGRKKRRQRRR	4.50 ± 0.23
R9	Cationic	RRRRRRRRR	5.26 ± 0.48
Transportan	Amphiphilic	GWTLNSAGYLLGKINLKALAALAKKIL	12.01 ± 0.18
MAP	Amphiphilic	KLALKLALKALKAALKLA	0.92 ± 0.06
PFV	Hydrophobic	PFVYLI	0.95 ± 0.05
Angiopep-2	Hydrophobic	TFFYGGSRGKRNNFKTEEY	0.91 ± 0.04
HAI	Hydrophobic	HAIYPRH	0.98 ± 0.03

Data presented here was mean ± s.d. of three independent experiments (*n* = 3). Median fluorescence intensity of CHO cells treated with the indicated peptide + bystander AgNPs was quantified by flow cytometry and normalized to that of cells with AgNPs alone. All the peptides used here were biotin-labeled ones. *AgNP* silver nanoparticle.

**Table 2 pharmaceutics-13-00552-t002:** Characterization of nanoparticles in this study.

Nanoparticles	Surface Coating	Z-Ave (nm)	PDI	Zeta Potential (mV)
AgNP-555	CF555, PEG2000	83.57 ± 0.16	0.100 ± 0.009	−26.3 ± 0.4
AgNP-647	CF647, PEG2000	100.40 ± 0.48	0.131 ± 0.005	−14.7 ± 0.6
AuNP	CF647, BSA	63.12 ± 0.47	0.253 ± 0.007	−30.2 ± 1.2
Au50	BSA	46.41 ± 1.31	0.306 ± 0.002	−11.9 ± 1.4
IONP	CF647, Dextran	57.30 ± 5.96	0.247 ± 0.090	2.7 ± 0.2
QD	CF647, PEG2000	32.93 ± 1.06	0.331 ± 0.069	−11.9 ± 1.4

Data presented here was mean ± s.d. of three independent experiments (*n* = 3). Zeta potential was measured at pH 7.4 for all NPs. *AgNP* silver nanoparticle, *AuNP* gold nanoparticle, *IONP* iron oxide nanoparticle, *QD* quantum dot.

## Data Availability

The data presented in this study are included in this published article.

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
