# Peer review of "Transportan Peptide Stimulates the Nanomaterial Internalization into Mammalian Cells in the Bystander Manner through Macropinocytosis"

_pharmaceutics, 2021, doi:10.3390/pharmaceutics13040552_

Round 1

Reviewer 1 Report

The authors did an excellent job addressing questions from the reviewers.  The paper is now impactful and well deserves publication.

“S” is missing from Figure S3 figure legend

Author Response

The title of Fig. S3 has been changed to "Dependence on scavenger receptors". 

Reviewer 2 Report

I am happy with the current version of the manuscript and suggest acceptance in its present form.

Author Response

Thank you.

Reviewer 3 Report

The manuscript (pharmaceutics-1175264) entitled "Transportan peptide stimulates the nanomaterial internalization into mammalian cells in the bystander manner through macropinocytosis" is well written with a reasonable set of experiments. This manuscript was previously submitted by the author with manuscript id - (pharmaceutics-1116688). The author has incorporated previously given suggestions but the inclusion of the following suggestions still lacking. In my opinion, the author has to consider the following suggestions especially considering the interest of the reader working in the area of drug delivery.  

1. It is suggested to include the future directions of the current investigation in the conclusion section in more detail especially considering the interest of the reader working in the area of drug delivery.

Author Response

We have added additional future directions in the conclusion section (highlighted in yellow).

This manuscript is a resubmission of an earlier submission. The following is a list of the peer review reports and author responses from that submission.

Round 1

Reviewer 1 Report

Summary

In this study, Li et al screened several known cell-penetrating peptides (CPP), and by testing them for AgNP uptake in CHO cells, they identified transportan (TP) to exert a considerable bystander effect that is stronger than TAT. The authors then characterized the activity of TP and its potential for drug-delivery in vitro using a panel of cell lines. They also tested this peptide with the slice of mouse tumors as an ex vivo model.

The study is designed with a clear goal, the experiments were straightforward, and the paper is written well. However, there are some important questions to be addressed.

Comments

  1. The authors examined TP internalization and bystander effect in tumor cell lines and fibroblasts. However, CPP must cross the blood vessel wall first to be useful for drug delivery. All studies (Fig. 1-5) should include experiments with cultured endothelial cells.

Is the TP effect influenced by the endothelial cell types (macrovascular or microvascular ECs, arterial, venous, capillary vessel ECs)?

  1. Table 1

The normalized AgNP uptake for TP is 12.01 and it is significantly higher than that of TAT. How is this compared with CendR peptides?

  1. It is an interesting finding that TP is a peptide with the bystander effect. However, does TP has a particular target site? How will it be useful for drug delivery if it doesn’t have a unique target? For instance, iRGD targets angiogenic vasculature (e.g. tumor vessels) via integrin and neuropilin binding without homing to normal quiescent blood vessels.

Related to this question, what is the molecular target for TP? The observation that TP bystander effect is dependent on HSPG and/or scavenger receptors doesn’t mean these are the direct molecular targets. The authors should discuss these remaining questions in the manuscript.

  1. What is the relationship between scavenger receptors and HSPG, functionally? The results suggest both are involved in the TP-induced internalization (micropinocytosis) mechanism.

  1. The authors used dextran sulfate to inhibit scavenger receptor function and argue that scavenger receptors are essential for the bystander effect. This conclusion needs to be supported by other methods, such as scavenger receptor knockdown experiments.

Likewise, the role of HSPG is examined solely by the addition of soluble heparin. The role of HSPG must be confirmed by knockdown of the genes required for HS synthesis such as EXT1/2. This is crucial because the result with CHO-pgs745 is only correlative.

  1. Fig. 4c,d

How do you know these are early/late endosomes without markers?  For this conclusion, you need a fluorescence NP study with endosome marker co-staining.

  1. Line 421 – The authors wrote “Our recent studies have instead defined a special micropinocytosis-like process initiated by ligand-receptor binding, which is crucial for cellular uptake of CPP-functionalized NPs and bystander ones”

However, the authors did not identify the receptors or the mechanism of internalization in this study. At this point, it is not known how the “special micropinocytosis-like process” is initiated by the cells. This statement needs to be toned down.

  1. General concerns regarding drug delivery using the bystander effect – With the bystander effect, there is no need for chemically conjugating drugs with a targeting moiety (peptide), and this is an advantage. However, the bystander effect will induce cells to uptake anything in the bloodstream such as plasma components as well as other drugs that the patient is taking, and this can be a major disadvantage. Elderly people are typically taking multiple medications. This would cause adverse effects at the treatment target site. Considering this unwanted effect, is the translational potential of the bystander effect actually realistic? How can this problem be circumvented? This is an important discussion point for the audience.

Reviewer 2 Report

The paper by Li et al. presents a thorough investigation into the effect of transportan (TP) peptide in promoting cellular uptake of a gamut of (mainly) metallic nanoparticles (NPs). Overall, I am satisfied with the manuscript. It is written nicely in lucid English while the data is well presented.

I think Figure 1 is confusing and at least Figure 1B is noway showing any bystander effect despite the fact that Figure 1C is making the point. Maybe the authors should consider also quantifying the fluorescence imagery datasets and quantify uptake also from there. I was also disappointed to see that the authors have not reported z-stacks despite doing confocal microscopy as a 3D rendition from z-stacks would have also given us an idea on whether the NPs, with or without TP, are internalized and localized intracellularly in a different or similar manner as that bit of information is important. 

I think the authors have not done justice to their effort as they can make a much stronger paper out of the data they actually have. For example, the actual DLS data in form of size distribution is missing whereas I think the hydrodynamic size of NPs is quite important and, in fact, can actually give a stronger explanation to some of the results reported in the manuscript. I invite the authors to take advantage of the supporting information section and release those data with arresting infographics and thorough analysis of the imagery datasets. That will augment the appeal of such a paper by manyfold.

Reviewer 3 Report

The manuscript (pharmaceutics-1116688) entitled "Transportan peptide stimulates the nanomaterial internalization into mammalian cells in the bystander manner through macropinocytosis" is well written with a reasonable set of experiments. Indeed, the manuscript required revision considering the following suggestions to further improve the quality of the manuscript.

  1. It is suggested to include the clinical significance of the present approach with suitable citation of the reference in its support in the introduction section.
  2. Authors are advised to summarize the findings of other investigators following a similar line of research in the introduction section.  
  3. It is suggested to include the value of PDI (polydispersity index) of each type of nanoparticulate system mentioned in table 2 and highlight the impact of PDI in nanomaterial internalization into mammalian cells.
  4. Authors are suggested to include the future directions of the current investigation in the conclusion.